# Correction and Accuracy of PurpleAir PM_2.5_ Measurements for Extreme Wildfire Smoke

**DOI:** 10.3390/s22249669

**Published:** 2022-12-10

**Authors:** Karoline K. Barkjohn, Amara L. Holder, Samuel G. Frederick, Andrea L. Clements

**Affiliations:** 1US Environmental Protection Agency Office of Research and Development, Research Triangle Park, Durham, NC 27711, USA; 2Former ORAU Student Services Contractor, US Environmental Protection Agency Office of Research and Development, Research Triangle Park, Durham, NC 27711, USA; 3Currently Department of Atmospheric Sciences, University of Illinois Urbana-Champaign, Urbana, IL 61801, USA

**Keywords:** air sensor, PurpleAir, PM_2.5_, air quality index (AQI), correction, evaluation, wildfire smoke

## Abstract

PurpleAir particulate matter (PM) sensors are increasingly used in the United States and other countries for real-time air quality information, particularly during wildfire smoke episodes. Uncorrected PurpleAir data can be biased and may exhibit a nonlinear response at extreme smoke concentrations (>300 µg/m^3^). This bias and nonlinearity result in a disagreement with the traditional ambient monitoring network, leading to the public’s confusion during smoke episodes. These sensors must be evaluated during smoke-impacted times and then corrected for bias, to ensure that accurate data are reported. The nearby public PurpleAir sensor and monitor pairs were identified during the summer of 2020 and were used to supplement the data from collocated pairs to develop an extended U.S.-wide correction for high concentrations. We evaluated several correction schemes to identify an optimal correction, using the previously developed U.S.-wide correction, up to 300 µg/m^3^, transitioning to a quadradic fit above 400 µg/m^3^. The correction reduces the bias at each air quality index (AQI) breakpoint; most ambient collocations that were studied met the Environmental Protection Agency’s (EPA) performance targets (twelve of the thirteen ambient sensors met the EPA’s targets) and some smoke-impacted sites (5 out of 15 met the EPA’s performance targets in terms of the 1-h averages). This correction can also be used to improve the comparability of PurpleAir sensor data with regulatory-grade monitors when they are collectively analyzed or shown together on public information websites; the methods developed in this paper can also be used to correct future air-sensor types. The PurpleAir network is already filling in spatial and temporal gaps in the regulatory monitoring network and providing valuable air-quality information during smoke episodes.

## 1. Introduction

Wildfires are one of the largest sources of the fine particulate matter (particles of 2.5 µm and smaller, PM_2.5_) that is found in the atmosphere, contributing over 30% to the ambient PM_2.5_ concentrations across the U.S. [1] and severely degrading air quality in areas downwind from wildfire-prone regions [2]. The number of fires, the burned areas, and the duration of the fire season are increasing [3], leading to severe smoke impacts on air quality, as observed in California in 2020 [4,5]. Moreover, there was a reversal of long-term downward PM_2.5_ trends in the western United States, due to wildfire smoke [6]. Exposure to wildfire smoke has been linked to increased all-cause mortality and respiratory morbidity, with the very young, elderly, and individuals with respiratory disease being especially susceptible [7]. There are clear increases in adverse health outcomes being routinely observed, with increasing daily averaged PM_2.5_ concentrations of up to 200 µg/m^3^ [8]. However, smoke concentrations can be far higher, with some communities experiencing concentrations of above 800 µg/m^3^ for multiple hours over multiple days [9]. Additionally, wildland firefighters routinely work in areas with extremely high smoke concentrations, experiencing shift-average PM_4_ (particulate matter with a median diameter of 4 μm) concentrations of approximately 500 µg/m^3^ over an entire fire season [10].

In the United States, public health guidance to mitigate the impacts of smoke is based on the air quality index (AQI). The AQI is calculated from 24-h averages of PM_2.5_ concentrations, corresponding to the evidence for PM_2.5_ health impacts [11]. However, during wildfire smoke events, PM_2.5_ concentrations can change rapidly, depending on the meteorology, the dynamics of the fire, and the local topography. Large increases or decreases in PM_2.5_ concentration can be observed as the winds shift over the course of a day, setting up diurnal patterns of widely varying smoke concentrations. In the United States, the NowCast AQI was developed to address these dynamic conditions and is a weighted rolling average of hourly PM_2.5_ concentrations [12].

In addition to public health guidance, air quality data are used for the occupational exposure of outdoor workers in California, Washington, and Oregon. Many of these regulations are at PM_2.5_ AQI levels in the moderate to unhealthy for sensitive groups range (AQI = 69–201, PM_2.5_ concentration = 20.5–150.5 µg/m^3^). At these levels, the regulations require exposure controls (e.g., workers must move indoors or into vehicles with filtered air, move to an alternate location, or wear the respirators provided) [13,14,15]. In addition, the California Department of Industrial Relations (commonly known as Cal/OSHA) requires that once the PM_2.5_ AQI exceeds 500 (500 µg/m^3^), the employer must provide and must expect employees to wear respirators [13]. Given these occupational thresholds for risk-reduction action, it is important to have accurate measurements up to 500 µg/m^3^ to support occupational health decisions.

Large spatial gradients in PM_2.5_ values can occur due to the localized smoke plumes and complex topography in the mountainous environment wherein many fires occur [2]. Therefore, the nearest regulatory or temporary monitoring site—sometimes 50–100 miles away—may not accurately represent the local smoke concentrations. Lower-cost air sensors can be used to fill in some of these gaps.

In order for air-sensor data to be actionable, sensors must be evaluated and, in some cases, corrected to provide accurate data over a wide range of concentrations and environmental conditions [16,17,18]. To evaluate and improve the performance of PM_2.5_ air sensors, they are typically run alongside conventional PM_2.5_ air monitors, in order to compare the reported concentrations [17,19,20]. It is important to evaluate the range of pollutant and environmental conditions that the sensors will be used to monitor as performance can be variable over different concentration ranges and can be influenced by environmental conditions, including relative humidity (RH) and temperature [21]. Sensor evaluations compared against air monitors are critical to gain actionable PM_2.5_ data.

Over the past 5 years, the PurpleAir map has been increasingly used by the public, news media, and even some air quality agencies to fill in gaps in the monitoring network. Previous work has shown that PurpleAir PM_2.5_ is highly correlated with the reference instruments, but it often reports higher values, in some cases by a factor of two, when compared to the reference measurements [17,22,23,24,25,26,27,28]. The PurpleAir response has shown some dependence on particle composition or size, with varying corrections needed for dust and light-absorbing aerosols [25,26,29]. However, a single correction factor has been shown to provide accurate PM_2.5_ data (RMSE = 3 µg/m^3^) across the United States, under a variety of aerosol types and environmental conditions [17]. The PurpleAir sensor exhibits a linear response to wildfire smoke at concentrations as high as 200 µg/m^3^ [23,30]. Laboratory studies at higher concentrations of simulated smoke and other types of PM have shown that at elevated PM_2.5_ concentrations, the PurpleAir has a quadratic or polynomial response up to 10 mg/m^3^ [24]. This nonlinearity may be due to a variety of reasons, including sensor design and data-processing algorithms [21]. To our knowledge, there have yet to be any evaluations of PurpleAir sensors that account for nonlinearity above 200 µg/m^3^ for wildfire smoke in the field. 

With an appropriate correction, the PurpleAir sensor has the potential to provide accurate PM_2.5_ concentrations and greatly expand our knowledge of the air quality impacts of wildfire smoke. In August 2020, publicly available PurpleAir sensor data, with correction and quality checks, were displayed as a pilot effort, shown on the United States Environmental Protection Agency’s (U.S. EPA) AirNow fire and smoke map (fire.airnow.gov). These sensors supplemented the relatively few PM_2.5_ monitors in the western United States, where most U.S. wildfires occur. As of January 2022, more than 12,000 PurpleAir sensors are shown on the fire and smoke map. Extreme smoke episodes over many parts of the western United States since 2020 have highlighted the need for more accurate PurpleAir data for very high concentrations of smoke (>200 µg/m^3^). 

This study describes the development and evaluation of an extension to the U.S.-wide PurpleAir correction for high concentrations due to wildfire. We used the collocated and nearby sensor monitor pairs throughout the country during both typical ambient and smoke-impacted conditions to develop a correction between the PurpleAir-estimated PM_2.5_ and the monitor PM_2.5_.

## 2. Materials and Methods

### 2.1. PurpleAir Sensors

The PurpleAir PA-II sensor is a lower-price (USD 230–260) PM sensor that is widely used globally, with many thousands of publicly reporting sensors. These sensors are purchased and deployed by individuals, community groups, government agencies, and others to better understand local air quality. The PurpleAir consists of two Plantower PMS5003 laser-scattering particle sensors, reporting PM_1_, PM_2.5_, PM_10_, and particle counts in 6 bins from 0.3 to 10 µm. The two Plantower sensors within the device (channels A and B) sample at alternating 10-s intervals and provide 2-min-averaged data (80-s averaged data prior to 30 May 2019); this provides a redundant measurement of PM that can be used for quality control. The sensors also include a BOSCH BME280 sensor reporting temperature and relative humidity (RH). This temperature and RH sensor is positioned above the PM sensors inside the PVC cap, resulting in measurements that are strongly correlated with ambient temperature and RH but are somewhat warmer (2.7 to 5.3 °C) and dryer (9.7% to 24.3%) than ambient conditions [23,31,32]. A Wi-Fi-enabled chip allows the data to be uploaded to the cloud in near-real time. PurpleAir also makes a PA-II-SD model that saves the data to an SD card for applications where Wi-Fi is not available or for users who prefer to have a backup copy of the data in case the Wi-Fi drops out.

For this project, the 2-min or 80-s averaged data were downloaded for online sensors from the ThingSpeak API, using Microsoft PowerShell. These data were saved as .csv files that were then processed and analyzed in R [33] and Python. The offline sensor data were manually downloaded from the SD cards. The 2-min or 80-s data were averaged up to hourly data (where data that were collected between 08:00 and 08:59 became the 08:00 average). Hourly averages were excluded that did not contain at least 75% of the expected points (i.e., 33 points for 80-s averaged data, 22 points for 2-min averaged data) [34]. The 24-h averages were then computed from the 1-h averages from midnight to midnight. Those 24-h averages that were less than 75% complete (i.e., 18 h) were excluded.

The PM data reported by PurpleAir, at the time of publication, are identical to the Plantower PMS5003 output and are reported with two different labels. The [cf = 1] data are identical to the [cf = atm] data below roughly 25 µg/m^3^, as reported by the sensor, and then, the [cf = 1] transitions from 25–100 µg/m^3^ to reporting 50% higher values at concentrations above 100 µg/m^3^, compared to the [cf = atm] channel [17]. Some previous correction methods correct the [cf = 1] data [17,22,28,35], while others correct the [cf = atm] data [25]. These labels were switched (i.e., the [cf = atm] label on the higher data column, now labeled [cf = 1]) by PurpleAir until late 2019, leading to some confusion and discrepancies in the older literature [31]. We used the [cf = 1] data in this paper since we, and others, have previously shown that it is more strongly correlated to reference monitors over the full range of concentrations [17,28].

A quality control procedure was applied to the hourly data to remove potentially spurious readings. The hourly data were considered valid if the difference between the A and B channel PurpleAir PM_2.5_ values were within 5 µg/m^3^, or the relative percentage difference was less than 70%, resulting in 2.7% of the hourly averages being removed. The percentage criterion was relaxed from the 62% thresholds used in previous analyses of 24-h averaged data [17], due to the sensor having higher noise levels at hourly averages.

### 2.2. Monitors

For this project, twenty-five sensors collocated or near-collocated with the monitors were identified across the U.S. These sites span eight or nine continental U.S. climate regions [36,37], with most of the smoke-impacted sites being located in the west and northwest.

#### 2.2.1. Collocated Monitoring in Typical Ambient Conditions

Many ambient sensors were part of the U.S. EPA’s long-term performance project (LTPP); PurpleAir sensors were collocated at the regulatory monitoring sites in: Research Triangle Park, NC; Wilmington, DE; Decatur, GA; Edmond, OK; Denver, CO; Phoenix, AZ; Missoula, MT; and Sarasota, FL, beginning around July 2019. Additionally, a few collocated sensors operated by local air-monitoring agencies in Appleton, WI, Atascadero, CA, Cedar Rapids, IA, Marysville, WA, and Topeka, KS were used (Table 1). These sites represent a variety of climates and experience variations in particle concentration, composition, and size. These data are representative of typical ambient conditions, as observed across much of the United States. 

**Table 1 sensors-22-09669-t001:** Ambient summary of the hourly averaged typical ambient dataset locations, comparison monitor types, date ranges, and the number of data points (N). T640 and T640x data over 35 µg/m^3^ are excluded from this table and the analysis in the paper. All ambient sensors were collocated by the EPA or a partner.

Location	Monitor Type	Date Range	N
Appleton, WI	BAM-1020	01/01/2019–24/01/2019	574
	T640	25/01/2019–09/01/2020	8171
Atascadero, CA	BAM-1020	01/01/2018–24/10/2019	15,108
Cedar Rapids, IA	BAM-1020	05/05/2018–08/10/2020	27,456
	T640	04/09/2019–06/08/2020	232
Decatur, GA	T640x	01/08/2019–31/08/2020	6868
Denver, CO	T640	14/08/2019–30/09/2020	8828
Edmond, OK	T640	02/08/2019–31/12/2019	3130
	T640x	03/01/2020–30/09/2020	5044
Marysville, WA	BAM1020	01/01/2019–31/08/2020	8900
	1405-F	25/10/2018–31/12/2018	1616
Missoula, MT *	BAM-1020	22/11/2019–28/07/2020	1738
Phoenix, AZ	TEOM	28/10/2019–31/07/2020	3652
RTP, NC *	T640x	01/08/2019–19/11/2019	1921
Sarasota, FL	T640	30/05/2019–30/06/2020	9145
Topeka, KS	T640	12/03/2019–30/06/2020	8288
Wilmington, DE	T640	27/07/2019–30/06/2020	5705
All Ambient	13 sensors ***	01/01/2018–08/10/2020	116,376
All Smoke and Ambient sites **	27 sensors	01/01/2018–20/10/2020	134,458

* A different sensor from that deployed during the smoke impact tests. ** Includes all the ambient sensors listed in this table and all the smoke sensors listed in Table 2. *** 13 sensors paired with 17 monitors as some sites ran multiple monitors simultaneously or switched monitor types during the collocation.

**Table 2 sensors-22-09669-t002:** Smoke-impact summary of the hourly averaged smoke-impacted dataset locations, monitor types, fires, date ranges, number of data points (N), concentration, and RH range. All PurpleAir sensors have >99% of the RH data present. Some sensors were collocated, while others were nearby sensor monitor pairs identified on the fire and smoke map. Most datasets are 2–3 months long, including both the smoke-impacted time and some typical ambient conditions.

Nearby or Collocation	Location	Monitor Type	Fire	Date Range	N
Smoke nearby identified on fire and smoke map	Bend, OR	M903 Nephelometry	Beachie Creek wildfire	01/08/2020–19/10/2020	1758
	Boise, ID	BAM-1020	Aged OR smoke	01/08/2020–20/10/2020	1753
	El Portal, CA	E-BAM	Creek Fire	20/08/2020–19/10/2020	1279
	Forks of Salmon, CA	E-BAM	Red Salmon Complex wildfire	14/08/2020–20/10/2020	1199
	Hoopa, CA	E-BAM	Red Salmon Complex wildfire	31/07/2020–20/10/2020	1632
	Keeler, CA	TEOM	Creek wildfire	01/08/2020–20/10/2020	1876
	Oakridge, OR	BAM-1022	Archie Creek wildfire	01/08/2020–19/10/2020	1865
	Oroville, CA	E-BAM	North Complex wildfire	26/08/2020–15/10/2020	1016
	Tulelake, CA	E-BAM	Red Salmon Complex and Slater wildfires	31/07/2020–20/10/2020	1720
	Atascadero, CA	BAM-1020	River—Dolan wildfire and ambient	01/08/2020–19/10/2020	1850
Smoke collocated by EPA or partner	Happy Camp, CA	E-BAM	Natchez wildfire	11/08/2018–29/08/2018	348
	Missoula, MT *	BAM-1020	Prescribed	18/07/2019–12/09/2019	1152
	Oakley, UT	E-Sampler	Prescribed	01/11/2019–04/11/2019	48
	Pinehurst, CA	BAM1020	Prescribed fire and aged wildfire smoke	20/10/2018–27/10/2018	157
	RTP, NC *	Grimm	Prescribed	13/03/2019–31/03/2019	429
All Smoke	15 sensors			11/08/2018–20/10/2020	18,082

* This is a different sensor from the sensor deployed during ambient testing.

#### 2.2.2. Collocated Monitoring in Smoke-Impacted Conditions

The smoke-impacted datasets were collocations of the PurpleAir sensors at ambient monitoring sites that were impacted by wildfire smoke and at temporary smoke-monitoring installations near prescribed fires or wildfires (Table 2 and Appendix A). A short-term collocation in Pinehurst, CA with a Met One BAM 1020, a federal-equivalent method (FEM) device, recorded elevated concentrations due to prescribed fire and aged wildfire smoke during October 2018. A long-term collocation with a Grimm EDM-180 (FEM) at the U.S. EPA’s Ambient Air Innovation Research Site (AIRS) in Research Triangle Park, North Carolina was impacted by the smoke from multiple prescribed fires. A long-term collocation with a BAM 1020 (FEM) in Missoula, Montana was impacted by a prescribed fire and smoke event. Two other smoke-impacted datasets were derived from temporary deployments located near fires. The temporary deployments were carried out near the 2018 Natchez wildfire in Happy Camp, CA, collocated with a Met One E-BAM (non-FEM) and a 2019 prescribed fire/pile burn near Oakley, UT, collocated with a Met One E-Sampler (non-FEM). 

#### 2.2.3. Nearby Sensor Monitor Pairs Identified on the AirNow Fire and Smoke Map

In addition to the true collocations performed as part of the U.S. EPA’s research projects, we identified sites with sensors located close to temporary or permanent monitors on the AirNow Fire and Smoke Map that were impacted by smoke from wildfires during the period of August to October 2020. Sites were selected that experienced smoke concentrations above 250 µg/m^3^, as measured by the monitor, as well as in other geographic locations with lower concentrations and less data collected (e.g., Boise, ID) (Table 2). All sites were within roughly 3 km of the nearest monitor (although many were closer) to minimize the impact of spatial and temporal variations in smoke concentrations on the comparison between the sensor and the monitor (Appendix A). 

#### 2.2.4. Air Monitoring Equipment and Data Access

##### PM_2.5_ Monitors: Permanent Federal Equivalent Method (FEM) and Temporary Smoke Monitors

Permanent monitors were installed at stationary monitoring sites as part of federal or state regulatory networks, using a variety of measurement technologies and instrumentation. Most permanent monitors have an FEM designation [38,39] that is suitable for the regulatory monitoring of PM_2.5_ via 24-h averages. FEM monitors used in collocations for this study included light-scattering-based PM_2.5_ detection methods, including the T640 or T640x (Teledyne API, San Diego, CA, USA) and the EDM-180 (GRIMM Aerosol Technik, Ainring, Germany), beta-attenuation-based PM_2.5_ measurements (MetOne BAM 1020 or BAM 1022, Met One Instruments, Grants Pass, OR, USA), and tapered-element oscillating microbalance-based PM_2.5_ measurements (TEOM 1405, Thermo Fisher Scientific, Waltham, MA, USA). Nephelometer-based (Radiance Research, Seattle, WA, USA) permanent monitors without a FEM designation were also used in this study since they report to AirNow and typically offer adequate performance [40,41,42].

Temporary deployments of non-FEM, portable PM_2.5_ instruments are employed in areas prone to wildfires and where permanent monitors are sparse. The MetOne E-BAM and E-Sampler are the most common monitors used in these temporary deployments. Although these instruments do not have a FEM designation, they do have quality control procedures (e.g., zero checks, leak checks, and flow audits) and have been evaluated with smoke, both in the laboratory [26,43,44] and in the field [45], and provide accurate PM_2.5_ smoke concentrations up to 1500 µg/m^3^.

Data from the permanent and temporary monitors were obtained using multiple different methods. Temporary monitor data were obtained through a direct data download from the instrument, via the Interagency Wildland Fire Air Quality Response Program (IWFAQRP) Monitoring 4.1 website (https://tools.airfire.org/monitoring/v4, accessed on 19 August 2021), and through the Airsis web-based instrument download (https://app.airsis.com/USFS/, accessed on 3 January 2022). The IWFAQRP data are quality controlled to remove data where the internal relative humidity is greater than 40%, the instrument flow rate is outside the range of 15.7–16.7 lpm, with negative PM_2.5_ concentrations set to zero. Any additional clear outliers were removed from the reference datasets (Appendix A). The direct instrument downloads do not have any quality-control procedures applied. Permanent data were downloaded from the U.S. EPA’s Air Quality System (AQS) and have been quality-controlled by local agencies. 

##### Quality Assurance of Permanent Monitors

The accuracy of the permanent monitors was evaluated using the PM_2.5_ continuous monitor comparability assessment tool, which allows the automatic plotting of the FEM data against the collocated federal reference method (FRM) filter data (https://www.epa.gov/outdoor-air-quality-data/pm25-continuous-monitor-comparability-assessments, accessed on 7 July 2022). Sixteen out of twenty-two monitors had comparability assessments; of these, eight of the sixteen monitors had slopes outside the target range (1 ± 0.1) (Appendix A). Many of these devices were Teledyne T640s and T640xs. T640s and T640x are optically based FEMs that may have less noise at low concentrations but may have nonlinear bias compared to other PM_2.5_ measurement methods [46] (Appendix A). Because of the nonlinear performance at high concentrations, T640 and T640x data above 35 µg/m^3^ was excluded and no T640 or T640x monitors were used as reference measurements during smoke events (see additional details in Appendix A). Data were used from T640 and T640x collocations at typical ambient sites up to 35 µg/m^3^ to build and evaluate the correction; these provide valuable data with low noise at low concentrations across more parts of the country than would have been represented if no T640 or T640x data were included.

### 2.3. Correction Development 

Most corrections for PurpleAir sensors do not account for the nonlinearity seen under extreme concentrations (Figure 1). We considered several corrections, including terms to account for RH (Table 3) since previous studies have suggested that humidity levels can impact the relationship between PurpleAir (or Plantower) and the monitor PM_2.5_ values [32,47,48,49]. Many of the equations considered in this paper include interaction terms (i.e., PM × RH) because RH and PM_2.5_ are slightly (but often significantly) correlated. We also considered a nonlinear impact of RH that was similar to that used in previous work [28]. 

#### 2.3.1. Withholding

To ensure that the selected correction models were not overfitted, we built and tested each model by withholding a portion of the dataset for evaluation only. We first built each model in Table 3 using all but one sensor and then tested it on the excluded sensor (“leave one sensor out”). This resulted in 27 iterations of each model type, with datasets from each sensor. Each sensor had a variable range of PM_2.5_ concentrations, but only five sensors reported data with concentrations above 500 µg/m^3^. The sensors also experienced a range of RH (0–100%), with some having typically low RH (e.g., the Phoenix mean has an RH = 18%), some having typically high RH (e.g., the Marysville mean has an RH = 74%), and some a broad range (e.g., the Topeka mean = 50%, range = 10–80%). The datasets also covered a wide range of times, with some sites representing longer time periods than others (Figure 2). We also withheld according to the week of the year, building on all but one week of the year (“leave one week out”), and then testing on that week. Many weeks had data from 3 years (i.e., 2018, 2019, or 2020) (see Appendix A). Most weeks had a wide range of RH conditions, while multiple weeks had high concentrations from smoke impacts.

#### 2.3.2. Correction Selection

We then evaluated the models by normalized mean bias error (NMBE) and simplicity. Simplicity was defined as the number of terms in the equation, with a linear model more simple than a quadradic model, a quadradic model more simple than a cubic model, and an RH term, increasing the complexity of each model type. If two models performed adequately, the simpler of the two models would be selected. We wanted to select a model that would keep the normalized mean bias error within 10% at each AQI breakpoint and at the threshold of 500 µg/m^3^, using a piecewise model, if needed, and employing the simplest model that would meet this requirement. We evaluated the performance at each AQI breakpoint (Appendix A) according to public health guidance changes at each level (e.g., limiting outdoor activity); therefore, this is where sensor accuracy is most important. We also evaluated the performance at 500 µg/m^3^, which is the Cal/OSHA level at which a respirator should be worn. We evaluated the NMBE at each breakpoint, using bins of data within ±20% around each breakpoint, to ensure an adequate number of hourly measurements in each bin. We also included bins for concentrations below and above the breakpoints to ensure results outside our targeted range were not unreasonable.

### 2.4. Correction Evaluation

After a correction was selected, the corrected dataset (generated using withholding) was evaluated at hourly, 24-h, and NowCast averages using the recommendations from the EPA air sensor performance target report for PM_2.5_ [50]. The report recommends procedures for evaluating sensors in field and lab environments and specifies metrics including the ordinary least squared regression slope, intercept, R^2^, root mean squared error (RMSE), and normalized root mean squared error (NRMSE), along with target thresholds. To meet the targets, the sensor must meet either the RMSE target or the NRMSE target; the NRMSE is typically used for high concentrations (e.g., smoke events), where larger absolute (RMSE) errors are common. Although the report recommends that sensors be evaluated at 24-h averages, the same criteria can also be used on 1-h averages. 

We also evaluated how often the corrected PurpleAir data reported the same NowCast AQI category as the monitor. The NowCast is a 12-h weighted average that is weighted based on the variability of concentrations (see details on the NowCast in the Appendix A). To convert the concentration into AQI, the U.S. EPA’s AQS reference table was used [51] (see Appendix A). Evaluating performance in both hourly and NowCast averages provides a better understanding of the accuracy of the data displayed on the fire and smoke map, which currently displays NowCast, hourly averages, and 10-min-averaged sensor data. The evaluation of 24-h averages allows us to better compare the corrected PurpleAir measurements to the performance targets and removes some of the uncertainty in the 1-h monitor measurements.

## 3. Results and Discussion

### 3.1. Selecting an Equation

#### 3.1.1. Collinearity of Terms

Many of the equations considered in this paper include the interaction terms (i.e., PM × RH) because RH and PM_2.5_ are slightly (but often significantly) correlated (PurpleAir [cf = 1] PM_2.5_ with PurpleAir RH R^2^ = 0.002, with collinearity of *p* < 2.2 × 10^−16^, Monitor PM_2.5_ with PurpleAir RH R^2^ = 0.003, *p* < 2.2 × 10^−16^). However, this requires a wider range of combinations of RH and PM_2.5_ (e.g., a high RH and high PM_2.5_, a high RH and low PM_2.5_, a low RH and low PM_2.5_, and a low RH and high PM_2.5_).

#### 3.1.2. Best Equation, Based on NMBE and Simplicity

The optimal equations were selected based on the NMBE at each breakpoint (Appendix A). No single proposed equation kept the NMBE at ≤10% across all breakpoints. The U.S.-wide correction equation was the best-fitting model for PM_2.5_ concentrations < 10 µg/m^3^; the NMBE is ≤10%, up to (and including) the breakpoint to the classification of “Hazardous” (250.4 µg/m^3^). At the Cal/OSHA limit, the U.S.-wide correction leads to an NMBE > 10%, and the quadradic fit is the simplest fit that keeps the NMBE at ≤10%. The results are similar whether using week or site withholding. Using a piecewise equation that includes both the U.S.-wide correction and the quadradic fit at higher concentrations meets the criteria of NMBE ≤ 10% at each breakpoint of interest (see Table 4).

#### 3.1.3. Final Equations

The U.S.-wide equation reduced the NMBE to ≤10%, up to the breakpoint to “Hazardous”, and the quadradic fit is the best and simplest fit around the Cal/OSHA respirator limit, based on the NMBE (best fit) and the number of terms (simplicity). Since the transition falls in the “Hazardous” category, we transitioned from roughly 300–400 µg/m^3^, corrected so that the transition zone does not impact data near either the “Hazardous” breakpoint or the Cal/OSHA respirator limit, keeping the error low throughout the transition (Appendix A). We transitioned linearly from using 100% of Equation (1) (model A, the original U.S.-wide correction) to 100% of Equation (3) (model B, with a quadradic fit), as shown in Equation (2) (Appendix A). The transition zone was selected by calculating the raw PurpleAir [cf = 1] (PA_cf1_) value that would result from a corrected PM_2.5_ concentration of 300 µg/m^3^, using the U.S.-wide correction at 50% RH and 400 µg/m^3^, employing the quadradic fit:
PAcf1 < 570 (corrected = 300 µg/m^3^ at 50% RH)
PM_2.5_ = PA_cf1_ × 0.524 − 0.0862 × RH + 5.75
(1)

570 ≤ PA_cf1_ < 611
PM_2.5_ = (0.0244 × PA_cf1_ − 13.9) × [Equation (3)] + (1 − (0.0244 × PA_cf1_ − 13.9)) × [Equation (1)]
(2)

PA_cf1_ ≥ 611 (corrected 400 µg/m^3^)
PM_2.5_ = PA_cf1_^2^ × 4.21 × 10^−4^ + PA_cf1_ × 0.392 +3.44
(3)


In the equations, RH represents the relative humidity, as measured by the PurpleAir.

The transition is needed to reduce the gap between equations. Simplicity is prioritized to reduce the computation time, which precludes the use of a sigmoid transition (i.e., a mathematical function with an S-shaped curve). We considered merging the datasets at the intersection between the two curves at 50% RH. However, since only the linear correction includes RH, this would lead to a discontinuity of up to 4 µg/m^3^ at high and low RH. For example, if the concentration rose by 1 µg/m^3^ at the transition point between equations, the estimated PM_2.5_ could fall by 4 µg/m^3^ if the RH was 0%. Therefore, the transition is effective in tapering the influence of the RH term. The transition is not essential; therefore, for applications that prioritize simplicity, Equation 1 can be used for concentrations up to 300 µg/m^3^, while Equation (3) can be used for concentrations of 300 µg/m^3^ and above.

#### 3.1.4. Discussion Influence of RH

As reported in our previous work [17], here, we do not see exponential increases in PM_2.5_ estimates at high humidity (>60–80% RH). Although the hygroscopic growth of particles can lead to exponential light scattering [52], it does not appear to impact the PurpleAir data used in this study, which is potentially due to the sensors keeping the particles slightly warmer or dryer, or is due to some particles growing to a size that is not easily detected by the PurpleAir [53]. The additive RH term is still effective at correcting and improving the sensor accuracy from 0 to 300 µg/m^3^.

It is unclear what the influence of RH may be on PurpleAir performance at high concentrations. Over ~750 ug/m^3^, the higher RH appears to lead to lower PM_2.5_ estimates from the PurpleAir sensors (Figure 1). However, this data is from only three sensors (Forks of Salmon, Hoopa, and Oroville, Appendix A). The RH range experienced at each of these sites during these high concentration times is quite limited (range < 24%), making it unclear whether this apparent RH influence is due to the difference in monitor sensor agreement at the 3 sites or is due to higher RH values, leading to sensor saturation at lower concentrations. Additional data would be needed to better understand the influence of RH at high PM_2.5_ concentrations; however, monitors may also demonstrate large uncertainties at these extreme conditions, which may lead to further challenges in quantifying this relationship.

### 3.2. Evaluation of the Correction—EPA Performance Target Evaluation at 1-h and 24-h Averages

Both the withholding methods resulted in similar sensor performance. The ambient sensors performed well, meeting or very nearly meeting the targets at both 1-h and 24-h averages (see details in Appendix A), with slopes within 1 ± 0.31, intercepts within ±2 µg/m^3^, RMSE ≤ 6 µg/m^3^, and R^2^ ≥ 0.66 (target R^2^ ≥ 0.70). Smoke-impacted sensors typically met or were near to the 24-h averaged targets with slopes within 1 ± 0.38 (target 1 ± 0.35), intercepts within ± 13 µg/m^3^ (target ±5 µg/m^3^), NRMSE ≤ 40% (target ≤ 30%), and R^2^ ≥ 0.61 (target R^2^ ≥ 0.70). At the 1-h averages, the smoke-impacted sensors still typically met or nearly met the targets, although in some cases, this was with larger errors (NRMSE of up to 55%). While not all the sensors evaluated met the performance targets, it is important to note that the performance targets are not pass/fail certifications but are instead a recommendation to encourage overall performance improvement of the technology entering the market. Falling short of these targets does not mean that the data are unusable. 

It is also important to note that not all recommendations in the performance targets were followed in this evaluation. Many of the smoke-impacted sites were selected as nearby monitor sensor pairs on the fire and smoke map and, therefore, may not be true collocations. Some of the scatter (RMSE) in the comparisons may be due to real concentration differences. In addition, the performance targets report recommends evaluating the sensors in triplicate. Here, we have evaluated a single sensor at each site. However, past work with PurpleAir sensors has already established their high precision (especially when using the duplicate Plantower sensors for quality control) and so this may be less of an issue for this sensor type [17,20,31,54]. Lastly, not all sensors were evaluated against FEMs, as recommended in the performance targets document [50]. However, this recommendation is less relevant for smoke monitoring, wherein FEMs also show uncertainty since their designation involves field-testing at typical ambient, not smoke-impacted, conditions.

### 3.3. Negative Values at Hourly Averages

Monitoring sites typically retain negatives that are near zero to represent the baseline noise of the instrument. After correction using withholding, 1.5% of the PurpleAir data were negative, compared to 1.9% of the monitor data. The lower percentage of negatives from the PurpleAir sensors versus the monitors suggests that the PurpleAir sensors are not reporting an unreasonable number of negative values. On the AirNow fire and smoke map, all the negative values that are displayed are rounded to zero; however, in this work, negative values have been retained for the purposes of analysis. 

### 3.4. NowCast Averaged Performance

Overall, the corrected data predicted the NowCast AQI category correctly for 94% of the time (Figure 3 and Appendix A). When the PurpleAir and monitor reported different AQI categories, often (77% of the time), the concentrations reported were within 5 µg/m^3^ (e.g., the PurpleAir concentration is 10 µg/m^3^ and the monitor concentration is 13 µg/m^3^, leading to different AQI categories but with only a concentration difference of 3 µg/m^3^). In all categories, <1% of the points are of more than one category different. 

The “Hazardous” category has the highest percentage of points that underestimate the AQI category (12%). There is a slight underestimation at the breakpoint between “Very unhealthy” and “Hazardous” (NMBE = −5%); however. much of this underestimation is due to the large variability in agreement between the PurpleAir and the monitors at these high concentrations (see the scatter plot in Figure 1). This is also reflected in the increasing RMSE at the breakpoint between “Very unhealthy” and “Hazardous” (RMSE = 31 µg/m^3^). It is also important to note the more limited number of values in the hazardous AQI category (N = 969, 0.6% of the full dataset), meaning that these results may be less representative of a larger sample size. Additional hazardous AQI category data could enable better-tuned correction in the future.

### 3.5. Comparisons to Other Published Corrections

At concentrations of up to 300 µg/m^3^, the extended U.S.-wide correction is consistent with other corrections that are currently in use on the PurpleAir map (ALT cf = 3, Woodsmoke, AQandU, LRAPA, map.purpleair.com, accessed on 26 July 2022) (Figure 4). However, all other corrections that are in use on the PurpleAir map significantly underestimate the values at high concentrations (>300 µg/m^3^) (the Nilson equation is not an option on the PurpleAir map and is discussed in the next paragraph). The ALT [cf = 3] correction [55] may improve performance near the limit of detection, but it does not lead to improved performance in other concentration ranges. Moreover, the ALT [cf = 3] correction is impractical, requiring more data fields and additional calculations, which increases the computational requirements for large-scale mapping, as used in the AirNow fire and smoke map or other such applications. However, better quantification near the limit of detection may be more important for sensors that are used in very clean environments (e.g., indoors) and when seeking to calculate more accurate long-term trends.

For this paper, we also considered the Nilson equation [28]. This equation is used in a PM_2.5_ map of Canada, showing the FEM monitors, along with the PurpleAir and AQ Egg sensors (https://cyclone.unbc.ca/aqmap/, accessed on 26 July 2022). After the initial publication of this paper, a corrigendum was released for Nilson et al.’s 2022 correction work [28], revealing that they had incorrectly reported the label of the PM_2.5_ data they had used. Nilson et al. reported that they had used cf_1 data when, in fact, they had used cf_atm data. The Nilson correction as intended with cf_atm data is in line with other corrections used for low-concentration PurpleAir data (Figure 4). The Nilson correction using cf_atm performs similarly to our US-wide correction [17] at low concentrations (Figure 5A), but at high concentrations shows strong underestimations compared to the extended US-wide correction (Figure 5B). Therefore, the Nilson et al. correction is not appropriate for extreme smoke. Note that Figure 5 shows both equations plotted against the cf_1 data where the cf_1 and cf_atm data have a 1:1 relationship until cf_1 = 30 µg/m^3^ and then transition to a relationship of cf_atm/cf_1 = 0.66 at cf_1 = 80 µg/m^3^.

We fitted and tested a model of a similar form (Model F, Table 3) on the cf_1 dataset in this study, which resulted in significant overestimates between the breakpoint to unhealthy values for sensitive groups (UHSG) to the breakpoint to “Hazardous” AQI (28–300 µg/m^3^, NMBE = 42% to 63%, Appendix A). The equation developed on the dataset in this paper has slightly higher coefficients than the equation developed by Nilson, but the coefficients are within 10% (Appendix A). The slightly higher coefficients suggest a higher PurpleAir/monitor ratio in our dataset, which is likely due to our use of cf_1 data instead of the cf_atm data used in Nilson’s paper. In addition, the RH term is slightly stronger, suggesting slightly more hygroscopic growth in our dataset. The Nilson correction does not agree well with the correction that we have developed in this paper.

### 3.6. Limitations 

Because some of the PurpleAir sensors in this work were identified from the AirNow fire and smoke map, we do not have any certainty of the accuracy of their location. This may lead to additional errors since they might not be truly collocated. Some sensor owners may specifically identify their sensor location inaccurately, to address privacy concerns. Sensors that were actually collocated (within <10 m) may have seen better agreement. While 3 km is farther than we would typically consider for collocation (typically <0.1 km), the 3 sites with longer distances still had strong correlations (hourly R^2^ = 0.94–0.96), which was in line with those of the true collocations (hourly, R^2^ = 0.54–0.97).

PurpleAir and the internal sensors are known to underestimate the values during dust events [17,35,56,57,58], which may have contributed to some of the errors at dust-impacted sites in this study. Dust may be harder to sample with these sensors because of the large particle losses in the inlet due to the low flow fans, particle loss due to internal turns in the sensor, lower light scattering to the mass ratio for larger particles, and the assumed size distribution used to convert light scattering from the sensor into mass that cannot adjust if larger particles are present [53,59,60,61]. Future air-sensor hardware or correction methods that could account for dust and the changing size distributions could lead to improved accuracy and would be a valuable addition.

This work did not specifically consider sensor drift and aging. There is some uncertainty in the long-term performance of these sensors, although the comparison of the duplicate Plantower sensors gives us more confidence in the measurements over time. Previous work done on PurpleAir sensors in California has suggested they can provide useful data for three years or more without substantial drift [62]. As the PurpleAir network ages, additional work should be done to understand any impacts on accuracy over the full range of concentrations.

Relative humidity data were missing from a small portion of the dataset (1%). We did not account for the accuracy of the RH measurement in this work, although past work has shown that the PurpleAir internal RH measurement is strongly correlated with true temperature and RH and has high precision [23,31,32]. If an RH sensor fails, a default value of 50% or a local average RH can be used as a substitute.

## 4. Conclusions

Air sensors can provide additional data critical during smoke impacts. While the U.S.-wide correction developed previously improves the accuracy of hourly data up to 300 µg/m^3^, transitioning to a quadradic fit at higher concentrations keeps the NMBE to ≤10% at each AQI breakpoint and at 500 µg/m^3^ (the Cal/OSHA respirator limit). The correction developed in this paper can be used to improve the performance of PurpleAir sensors in situations from typical ambient concentrations to extreme smoke. The equation developed in this paper is not comparable to other commonly used PurpleAir corrections that are focused on a narrower range of concentrations.

This work is an evaluation of a single air-sensor type and there are many additional sensors commercially available. The framework from this study may be used for additional air sensor types. It could be used to both develop corrections and better understand sensor performance over the full range of anticipated concentrations, where decisions must be made (e.g., 0–500 µg/m^3^). Understanding sensor performance over a wide range of concentrations is critical for an application where sensors will provide information during both smoke-impacted and typical ambient conditions. The findings from this paper may also be relevant for other pollutant types, wherein understanding the performance over the full range of decision-relevant concentrations is important as well.

## Figures and Tables

**Figure 1 sensors-22-09669-f001:**
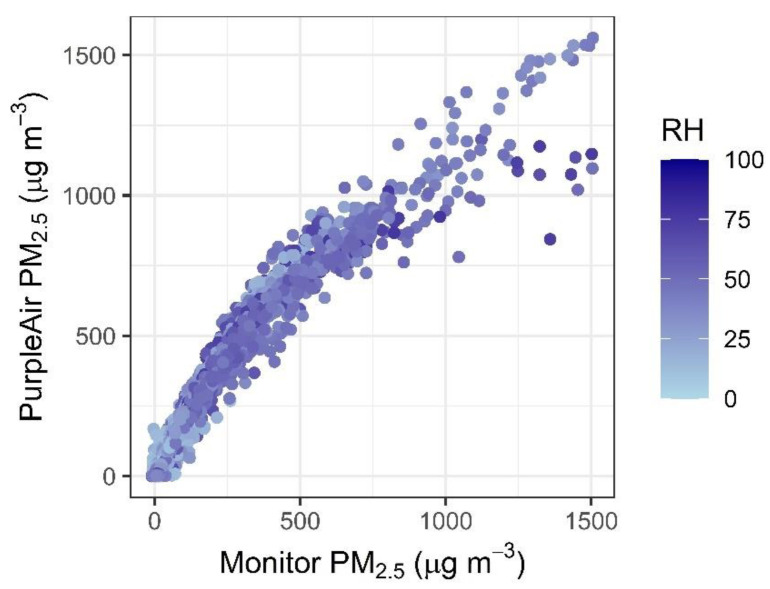
Nonlinearity in hourly PurpleAir PM_2.5_, uncorrected [cf = 1], versus the monitor PM_2.5_ values, colored by the relative humidity, as measured by the PurpleAir.

**Figure 2 sensors-22-09669-f002:**
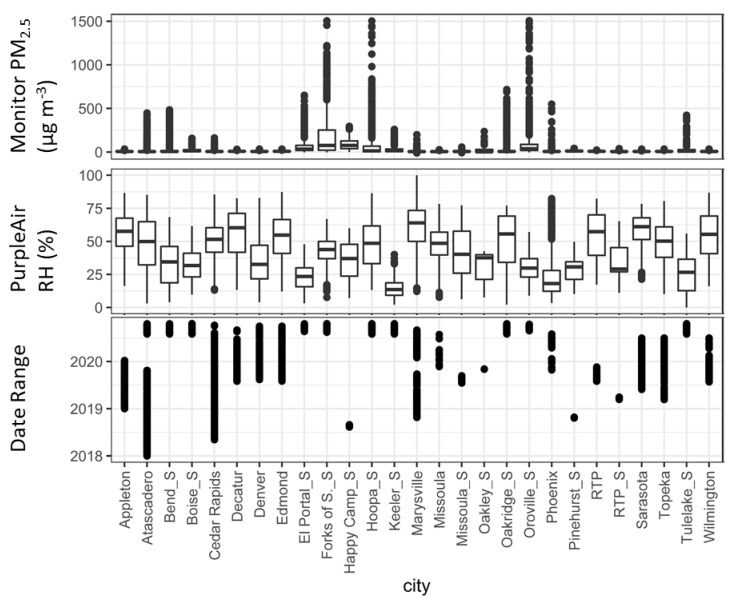
Summary of the distribution of RH and PM_2.5_ for each sensor (withholding groups). The Atascadero sensor experienced both typical ambient and smoke−impacted periods.

**Figure 3 sensors-22-09669-f003:**
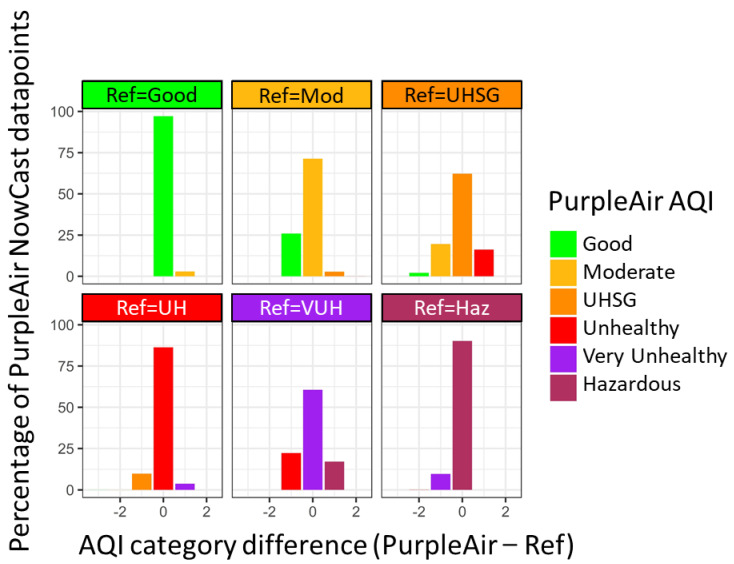
Percentage of the time that the PurpleAir NowCast AQI is the same as or different by 1 or 2 NowCast AQI categories. This is binned by the reference NowCast AQI category. It includes data using both withholding methods.

**Figure 4 sensors-22-09669-f004:**
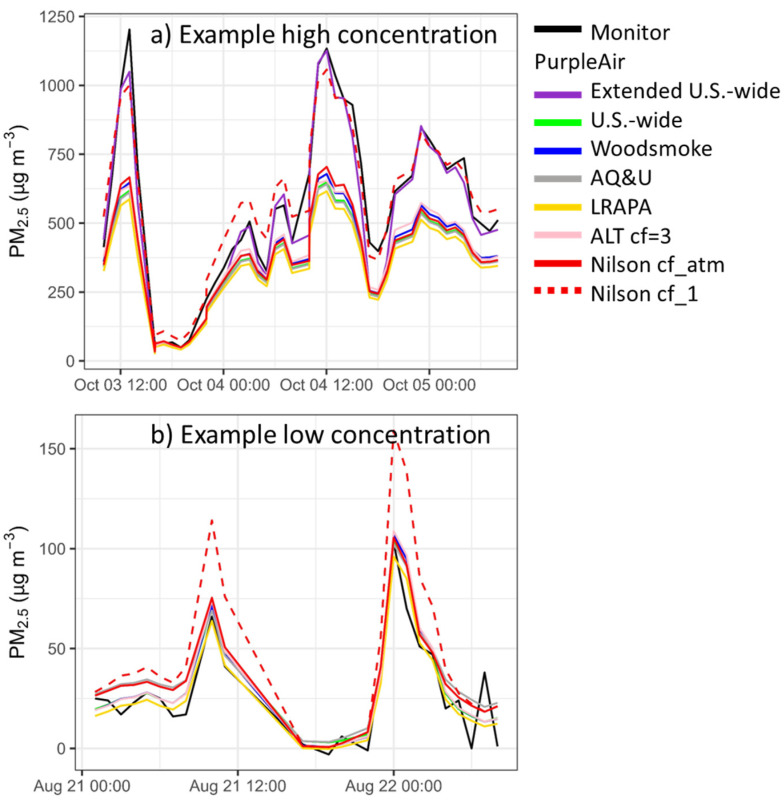
Example time series from the Forks of Salmon, CA, showing the data that were corrected based on different corrections on the PurpleAir map (Woodsmoke, AQandU, LRAPA, ALT cf = 3) and in the literature (Nilson cf_atm and cf_1).

**Figure 5 sensors-22-09669-f005:**
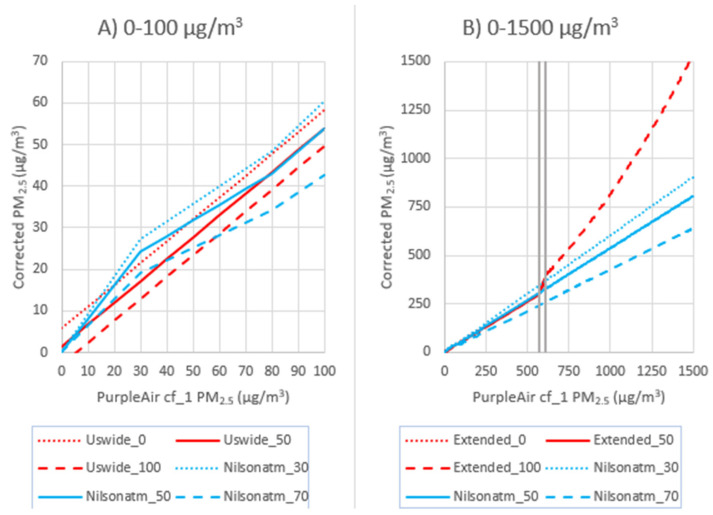
Disagreement between Nilson cf_atm (actually used in Nilson’s paper based on their corrigendum) and the extended US-wide correction, plotted over the range of RH used in each correction, shown at low concentrations (**A**) and over the full range (**B**) with gray vertical lines showing the transition zone from the US-wide correction to the quadradic fit.

**Table 3 sensors-22-09669-t003:** Correction equations that were considered.

Model	Fit	Equation
A	U.S. correction	PM_2.5_ = 0.524 × PA_cf_1_ − 0.0862 × RH + 5.75
B	Quadratic	PM_2.5_ = a × PA_cf_1_^2^ + b × PA_cf_1_ + c
C	Cubic	PM_2.5_ = a × PA_cf_1_^3^ + b × PA_cf_1_^2^ + c × PA_cf_1_ + d
D	Quadratic + RH	PM_2.5_ = a × PA_cf_1_^2^ + b × PA_cf_1_ + c + d × RH
E	Quadratic PM × RH	PM_2.5_ = a × PA_cf_1_^2^ + b × PA_cf_1_ +c + d × RH + e × PA_cf_1_ × RH
F	RH growth, Nilson [28] *	${\mathrm{PA}}_{\mathrm{cf}\_1}/\mathrm{monitor}=a+\frac{b}{\left(\frac{100}{RH}-1\right)}$ (RH limited 30–70%)

* After the initial publication of this paper, a corrigendum was released for Nilson et al.’s 2022 correction work [28], revealing that they had actually used cf_atm data. Cf_1 data have been used for Nilson data throughout the paper, except in some figures, where it is explicitly noted as Nilson atm.

**Table 4 sensors-22-09669-t004:** The 1-h error at each AQI breakpoint, using site withholding and a transition from the U.S.-wide correction to a quadradic fit from 300 to 400 µg/m^3^ (raw [cf = 1] 570–611 µg/m^3^).

AQI Breakpoint	Concentration Range (µg/m^3^)	N	Selected Fit	NMBE	RMSE (µg/m^3^)	NRMSE
Below	0–10	90,960	U.S.-wide	−5%	3	54%
Moderate	10–14	15,205	U.S.-wide	−10%	4	31%
UHSG	28–42	2196	U.S.-wide	4%	9	27%
Unhealthy	44–66	1291	U.S.-wide	9%	12	22%
Very Unhealthy	120–180	503	U.S.-wide	2%	18	12%
Hazardous	200–300	475	U.S.-wide	−5%	31	12%
Cal/OSHA	400–600	230	Quadradic	0%	89	18%
Beyond	600+	189	Quadradic	−7%	216	25%

## Data Availability

Data will be available after publication from DOI: 10.23719/1528138.

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
