# Peer review of "Correction and Accuracy of PurpleAir PM2.5 Measurements for Extreme Wildfire Smoke"

_sensors, 2022, doi:10.3390/s22249669_

Round 1
Reviewer 1 Report
This is a very nice manuscript that extends the correction of a low-cost AQ sensor data to the high concentrations found in wildfire and prescribed burn smoke. The PurpleAir sensors are compared to EPA reference sensors (FEM and some other high-quality sources) for both ambient PM and smoke-influenced PM. The results show that a previous correction is still valid at low PM values, but that a new fit is necessary to correct biases at high concentrations. One surprising finding is that the new correction for high PM does not depend on RH.
There are some errors to be corrected, but overall the manuscript is of good quality and the results are sound. The methodology section could be improved with an overview of the approach. While the approach is similar to previous efforts, Sensors has a wide readership and the authors should want their work to be more approachable to a more general audience. I have a number of specific comments below, but big picture the authors should consider more carefully the implications of not including RH in the new, high-PM correction.
Line 30: add “wildfire smoke” as a keyword.
Lines 48-49: you give the explicit definition of PM4 here, but don’t above on line 35 when you introduce PM2.5 While PM2.5 is close to becoming common knowledge, I would still move this definition to line 35.
Lines 62-75: this is all interesting and provides important motivation for your research. But providing all the cuts and specific numbers becomes tedious and unnecessarily lengthens the paper. Give one example and then simply note the other instances that rely on AQI without giving all the specifics.
Line 82 or 87: you already site Wallace (#48) later in the paper, but here you should also site Wallace 2022: Sensors 2022, 22(13), 4741; https://doi.org/10.3390/s22134741.
Line 111: either at the of Introduction or as a first paragraph in methodology, insert a short paragraph that gives the overall summary of methodology. On line 113, you jump right in and haven’t even discussed what you are using as the pair for the PAs.
Table 1: good info, but since you haven’t given the big picture, it’s difficult to interpret at this point. You sum up to 13 ‘sensors’, but this is sites since you have multiple sensors per site at several locations. Change the terminology.
Table 2: for N, not all measurements at the site are impacted by smoke, correct? The date range is often two months, and I assume the fire was active for only a small portion of the time. Need to give some additional information about this.
Line 148: define FEM
Line 158: be more clear: both the PA sensors and the ambient sensors were found from AirNow?
Line 164: confusing, since Boise and Missoula are listed in different parts of Table 2.
Line 168: The order is confusing, especially since there is no overview of methodology. You described the colocation first (2.1) and then the sensors (2.2). It would make more sense to swap the order. First, describe the sensors and how they work, and next discuss the colocation.
Line 184: be consistent with using data in the singular or plural, particularly within the same sentence. You are mostly using data in the singular.
Line 220-221: give a citation that this method would give equivalent results.
Figure 2 and Table 2: You have 15 smoke sensors listed in Table 2 but only 14 in Figure 2.
Lines 331-332: either use superscript for the exponential notation or add “e” and remove “x 10”.
Lines 349-359: include a simple figure that has this visually. Graph out the two equations, and also include the linear fit between the two. Ideally, you should have all of both equations, and then use color or line weight to show when each is valid. You could also put the fit on top of the data for Figure 1, if it’s not too busy. Yes, you’d either have to pick one RH or use some shading to indicate a range of RH values.
Lines 372-379: Looking at Figure 1, clearly RH has an effect on high concentrations. I understand that given your criteria of fit, you prioritize simplicity and therefore your procedure finds that including RH is not worth the extra complexity. But I am curious how your results would vary if you included the same RH term from the US correction (- 0.0862 x RH) in the quadratic equation. That would be akin to model D in Table 3, but you would not let the RH coefficient vary: instead it would be fixed from the results from the lower concentrations fit of Model A. Then you could have a smooth transition between the two fits and not need the linear interpolation between 300 and 400.
Lines 412-413: I agree with this point generally, but the conclusion of this paragraph is weak. Perhaps rephrase to “in this work negatives have been retained for analysis.”
Line 416: need period and space after parentheses.
Line 415-420: while overall this is a good confirmation of your approach, unfortunately the Ref=Haz skews in an unfavorable direction for the worst AQI: ~10% of the time, PurpleAir gives Very Unhealthy when the reference monitor is Hazardous. Please comment on this.
Lines 428-429: looks like Nilson is pretty good too at high concentrations. You mention this below, but this doesn’t read very smoothly right now. Perhaps you added Nilson later? In any case, need to make some comment at this point to correct this sentence.
Line 448: I think model F is in table 3, right? If so, site what’s in the paper, not in the supplementary material.
Line 456-457: you could make this case more strongly with a figure. You could include the Nilson fit on the same figure I suggested above. Again, you would need to pick some RH ranges for a 2-D (x,y) plot.
Reviewer 2 Report
After review, I recommend this manuscript can be accepted for publication in this status.
Author Response
We thank the reviewer for their time reviewing our manuscript.
Reviewer 3 Report
A good study of the opportunity to use low cost PM monitors to provide the temporal and spatial resolution needed to ensure healthy practice near wildfires. The analysis was clear and thorough. My main concerns were RH correction and limitations of the Plantower with larger particles.
I was expecting more discussion for the underlying reason the PurpleAir was not linear above 200ug/m3. This could have helped lead to a physically based choice of the correction algorithm above 50 ug/m3.
Your criterion for rejecting outliers as if >5 ug/m3 or 70% difference between the two channels. How often did this occur and was the 70%/5ug rule based on statistical analysis of the data?
I was expecting at least one graph of particle size distribution of the clean and wildfire areas. The 640 provides this information. The Plantower has limited response to particles >1.5um.
At RH >60% the particles grow exponentially. Hanel, Skupin, Koppen provide growth factor equations. A good reference is Hagan, D. H. and Kroll, J. H.: doi.org/10.5194/amt-13-6343-2020, 2020. You could separate your data according to %RH, then determine linearity for all data <60%RH, up to 500ug/m3. Then determine the additional high RH correction when applying your <60%RH correction equation to the high RH data.
Data statistics: the clean analysis is missing NRMSE and the smoke data is missing RMSE results.
Please state the BME 280 T and RH errors. It specifies +/-3%RH at 25C but does not specify temperature dependence of the 3%RH (maximum?) error. Temperature is +/-1C at 25C.
It is not surprising that the PurpleAir underestimates dust. If you compare a particle size distribution of clean or roadside air with wildfire or dust, the larger particles are not seen by the Plantower optics. Plantowers are calibrated for urban particles.
I believe you included negative values in your regression fitting. I think it is correct to include them, otherwise the intercept is skewed.
